# A Comparison of Non-Destructive Visceral Swab and Tissue Biopsy Sampling Methods for Genotyping-by-Sequencing in the Freshwater Mussel *Fusconaia askewi*

**DOI:** 10.3390/genes14061197

**Published:** 2023-05-30

**Authors:** Matthew Harrison, V. Alex Sotola, Alexander Zalmat, Kyle T. Sullivan, Bradley M. Littrell, Timothy H. Bonner, Noland H. Martin

**Affiliations:** 1Department of Biology, Texas State University, 601 University Drive, San Marcos, TX 78666, USA; matthar@gmail.com (M.H.); asz5@txstate.edu (A.Z.); tbonner@txstate.edu (T.H.B.); 2Biology Department, State University of New York at Oneonta, 108 Ravine Parkway, Oneonta, NY 13820, USA; 3BIO-WEST, Inc., 1405 United Drive, Suite 111, San Marcos, TX 78666, USA; ksullivan@bio-west.com (K.T.S.); blittrell@biowest.com (B.M.L.)

**Keywords:** non-destructive DNA sampling, DNA collection methods, Texas pigtoe, population genetic structure, genomic coverage, sequencing depth

## Abstract

Limiting harm to organisms caused by genetic sampling is an important consideration for rare species, and a number of non-destructive sampling techniques have been developed to address this issue in freshwater mussels. Two methods, visceral swabbing and tissue biopsies, have proven to be effective for DNA sampling, though it is unclear as to which method is preferable for genotyping-by-sequencing (GBS). Tissue biopsies may cause undue stress and damage to organisms, while visceral swabbing potentially reduces the chance of such harm. Our study compared the efficacy of these two DNA sampling methods for generating GBS data for the unionid freshwater mussel, the Texas pigtoe (*Fusconaia askewi*). Our results find both methods generate quality sequence data, though some considerations are in order. Tissue biopsies produced significantly higher DNA concentrations and larger numbers of reads when compared with swabs, though there was no significant association between starting DNA concentration and number of reads generated. Swabbing produced greater sequence depth (more reads per sequence), while tissue biopsies revealed greater coverage across the genome (at lower sequence depth). Patterns of genomic variation as characterized in principal component analyses were similar regardless of the sampling method, suggesting that the less invasive swabbing is a viable option for producing quality GBS data in these organisms.

## 1. Introduction

Studies examining the population genetic structure—or lack thereof—of rare and threatened species are critical tools for enabling biologists to recommend management strategies of these organisms [1,2,3,4,5]. Large sample sizes are often needed to obtain robust estimates of population genetic parameters, and it is important to minimize the degree to which genetic sampling results in the disruption or death of individual organisms, especially for rare species. A number of nondestructive sampling techniques (e.g., saliva, hair, swabs, etc.) have thus been developed to reduce the potentially harmful impact of genetic sampling [6,7,8,9,10].

In North America, the family Unionidae is a speciose group of freshwater mussels consisting of both widely distributed and endemic species that exhibit varying conservation statuses [11,12,13]. Dam construction, channel modification, siltation, and the introduction of non-native freshwater mollusks have all contributed to population declines in many of the species within this diverse group [13]. A diverse array of DNA collection methods has been proposed in order to reduce the mortality of sampled and re-released unionid mussels. Such methods include the clipping of small pieces of mantle or foot tissue (i.e., “tissue biopsies”), swabbing of the viscera, foot scraping, and hemolymph extraction via hypodermic needles [14,15,16]. These four methods were specifically evaluated for the freshwater pearl mussel (*Margaritifera margaritifera*) with respect to survivorship, growth rates, and DNA quantity and quality of the sampled individuals [16]. All four methods reliably produced DNA, though yields from hemolymph and foot scraping were significantly lower than those produced from tissue biopsies and swabbing. The authors additionally evaluated individual survivorship (after 128 days) across each of the sampling techniques and found that survivorship was 100% for all methods. Since the DNA quantity and quality of swab samples were not only reliable but also less invasive than tissue biopsies, the authors concluded that visceral swabbing was the most appropriate method for tissue collection in *M. margaritifera* [16]. A separate investigation in the pimpleback mussel, *Quadrula pustulosa*, further determined that tissue swabbing was the preferred method for DNA collection over mantle clipping because it produced high quality samples with less physical injury to the organisms [15]. 

Visceral swabbing methods have produced quality DNA appropriate for use in traditional Sanger sequencing methods. These have included the generation of ND-1 mitochondrial sequences [15] as well as microsatellite loci [16]. However, such methods do not require large amounts of targeted DNA to be extracted, and contamination with other non-target organisms is largely not problematic because primers are often designed to be species specific. The advent of genotyping-by-sequencing (GBS) methods, on the other hand, has made possible the routine production of orders of magnitude more genetic markers, including single nucleotide polymorphism (SNPs), at a fraction of the cost compared with traditional Sanger sequencing methods [17]. Massault et al. [18] evaluated the effectiveness of swabbing methods for GBS sequencing in the silver-lipped pearl oyster (*Pinctada maxima*, family: Margaritaferidae) and found such methods to be appropriate for parentage analysis and pedigree reconstruction in these marine bivalves. Collections via swabbing produced DNA of sufficient quantity and quality for genotyping, and SNP counts were also highly correlated among tissue and swab collection methods [18]. It remains unclear, however, whether DNA collection methods via swabbing are similarly appropriate for GBS sequencing of freshwater unionid mussels, and further investigation is needed to address this question.

In the current study, we evaluated the degree to which visceral swab and tissue biopsies generate high-quality GBS/SNP genotype data for the unionid freshwater mussel Texas pigtoe (*F. askewi*). This was achieved by collecting and processing swab and tissue biopsy samples from the same individuals and generating separate SNP datasets for both types of sampling methods, with swab samples being collected prior to performing tissue biopsies. Our primary goal was to determine whether the less-disruptive swab sampling method was sufficient to undertake large-scale population genetic studies using GBS methods for freshwater mussels. Therefore, we sought to (1) compare DNA concentrations and number of raw reads produced from visceral swab samples and tissue biopsies collected across 14 different individuals, (2) determine whether and how SNP coverage and numbers differed across the two sampling methods, and (3) determine whether the patterns of genetic variation detected among individuals are consistent across both datasets when assembled independently from one another, as well as when assembled to a larger tissue-based de novo reference assembly.

## 2. Materials and Methods

### 2.1. Sample Collection

A total of fourteen *F. askewi* mussels were collected via snorkeling and tactile searches at two separate locations along the Sabine River in Texas, USA, at 32.62986° North 32.62986° West (*n* = 6), and at 32.462220° North and 94.845864° West (*n* = 8). The mussels were identified in the field as *F. askewi*, on the basis of morphological characteristics, and were subsequently sampled using Isohelix SK-1S buccal swabs by carefully rubbing both sides of the swab along the mussel foot tissue of the mussel for 30 s. Swab heads were immediately placed in individual containment tubes and kept on dry ice until storage at −20 °C. After swabbing, a ~0.5 cm^3^ tissue plug was collected from the same individual using a nasal biopsy tool, and the mussel was then returned to its original location. Tissue samples were stored separately in 95% ethanol on dry ice before transferring to a −20 °C freezer for longer-term storage. 

### 2.2. DNA Extraction and Quantification, Library Preparation, and De Novo Assemblies

Whole genomic DNA was extracted from the 14 swab samples and the replicated 14 tissue biopsies using Qiagen DNeasy blood and tissue kits in a 96-well format. These samples were part of a larger group of 378 mussel tissue collections that were used in a separate population genetic study [19]. No attempts were made to equalize DNA concentrations prior to DNA library construction. However, DNA concentrations of the replicated swab and biopsied tissue samples were quantified post-hoc to investigate whether initial concentrations affected the number of reads per individual in the resulting libraries. DNA concentrations were determined using the Qubit^®^ dsDNA HS assay (Invitrogen; Waltham, MA, USA) kit following the standard protocols described in the Qubit manual. The DNA concentration of each of the 14 swab and tissue replicates was measured from a single aliquot three separate times, and a mean concentration was calculated for each sample. 

Extracted DNA from the broader collection of mussels (*n* = 378 [19]) was used to create a reduced-complexity genomic library. This library included the 14 tissue and 14 swab replicates utilized here and was generated following modified protocols commonly used in our laboratory group across a wide diversity of species [5,19,20,21,22]. Restriction enzymes EcoRI and MseI were used to digest extracted DNA; EcoRI adapters (i.e., 10–20 base-pair multiplex identifier sequences: MIDs) were ligated onto the resulting fragments, and the 14 swab and 14 tissue replicates were each assigned unique barcodes. Labeled products were amplified through two rounds of PCR using Illumina primers. PCR products were pooled into a single library and sent to the University of Texas Genomic Sequencing and Analysis Facility (Austin, TX, USA) where the samples were size selected for 300–400 base pair length fragments using BluePippen technology and sequenced on an Illumina Novaseq SR100 platform. 

A number of data processing steps were used to ensure high-quality sequencing data. First, Bowtie v.3 was used to identify PhiX sequences that are used in Illumina control libraries, and those reads that assembled to the PhiX genome were removed [23]. Custom Perl scripts (available from the corresponding authors) were used to match sample IDs with unique barcode identifiers as well as to remove Mse1 adapters and barcodes from sequence reads. The resulting sequence data, ranging from 84–86 base pairs in total length, were then organized into different assemblies for separate analyses. Since no reference genome is available for *F. askewi*, three separate de novo assemblies were built using different sets of individuals and collection methods. The first reference genome was created using a much larger dataset of tissue-sampled *F. askewi* individuals (*n* = 96, [19]). We used this reference to align reads for both tissue and swab replicates, enabling a direct comparison of resulting SNP datasets (i.e., to directly assess coverage, depth, and genetic variation of “tissue” and “swab” samples of the same 14 individuals). Two additional reference assemblies were also separately created: one exclusively using the 14 tissue samples (henceforth referred to as the “tissue-only” assembly), and another using the 14 replicated swab samples (henceforth referred to as the “swab-only” assembly). Each of the three separate reference assemblies (i.e., control reference, tissue-only assembly, and swab-only assembly) followed the same processes using part of the dDocent variant calling pipeline [24]. First, unique reads were identified for each individual, and reads with less than four copies and shared across fewer than four individuals were removed. Subsequently, the remaining sequences (i.e., those that had more than four copies and were shared across more than four individuals) were assembled to build each reference using CD-hit by utilizing a similarity threshold of 80% [25,26]. 

### 2.3. Number of SNPs and Coverage

Sequence reads were assembled to reference contigs using the aln and samse algorithms from BWA (version 0.7.13-r1126; [27]). BCFtools (version 1.9) was used to both identify SNPs as well as to calculate Bayesian genotype likelihoods for each SNP [28]. Biallelic SNPs must have been represented by at least 80% of the individuals in a given assembly and must have had a mean sequence depth of ≥2Xn (where *n* = number of individuals) in order to be included in the respective dataset. A custom Perl script was used to remove potentially paralogous loci with exceptionally high sequence depth (mean sequence depth > assembly-wide mean + 2 × sd). This script also serves to filter loci based on mapping quality (minimum score of 30), as well as the difference in base and mapping quality between the reference and alternative alleles using Mann–Whitney U tests (z-score cutoff = 1.96). Genotype likelihoods were assigned to each SNP for each individual and used to estimate allele frequencies; SNPs with a minor allele frequency of <0.05 were not included in the final datasets. If there was more than one SNP identified on an individual contig, only a single SNP was randomly chosen to include for analyses in order to reduce the effects of linkage disequilibrium.

### 2.4. Estimation of Genotype Probabilities

Rather than “calling” SNPs, we utilized the calculation of genotype probabilities which account for uncertainty associated with sequencing. Genotype probabilities were estimated using entropy, a hierarchical model that estimates Bayesian-based admixture proportions and genotype probabilities for each individual for a predetermined number of populations (k) [5,29]. Calculation of posterior distributions for k = 2 was carried out by combining two separate Markov Chain Monte Carlo runs that included 50,000 total iterations with a burn in of 5000 and sampling every 10th iteration. Genotype probabilities were averaged across both runs after checking for chain convergence with Gelman–Rubin diagnostic statistics.

### 2.5. Data Analysis

We performed a paired t-test to determine whether there was a significant difference between DNA concentrations that were collected for tissue and swab samples. We then used an analysis of covariance (ANCOVA) to assess the relationship between DNA concentrations and the total number of raw reads that were generated, while simultaneously accounting for sample type. This ANCOVA was also used to determine whether there was a significant difference in the number of reads that were produced by each sampling method. 

We ran principal components analysis (PCA) on genotype probabilities to determine whether SNP alignments to either the sample-type reference or the tissue control reference influenced the resulting patterns of genetic variation. In total, we ran four PCAs with: (1) swab data aligned to the swab-based de novo assembly, (2) tissue data aligned to the tissue-based de novo assembly, (3) swab data aligned to the control-reference-based de novo assembly, and (4) tissue data aligned to the control-reference-based de novo assembly. Rather than focusing on the exact genetic relationships that existed among individual samples, analyses were primarily focused on how relationships among individuals changed given the sampling type and the references to which they were aligned. To further examine this, we ran Procrustes analysis from the vegan package in Program R and reported the Procrustes R, a measure of correlation between two multivariate plots [30]. The first Procrustes analysis was run to compare the two PCAs, which were sample specific; swab and tissue data aligned to their specific de novo assemblies. The second Procrustes analysis compared the two sampling types when aligned to the control reference. We posited that if swab samples produced a SNP dataset with similar genetic patterns to the tissue samples, even when aligned to separate de novo assemblies, the PCAs would be highly correlated (>0.95). 

## 3. Results

### 3.1. DNA Concentrations

Mean DNA concentrations significantly differed among tissue and swab samples (two-tailed paired *t*-test, *t* = 2.171, *p* = 0.049, df = 13) with tissue samples (mean ± 1 S.D.; 36.2 ng/mL ± 14.6), having 38% greater DNA concentrations compared with those of swab samples (27.8 ng/mL ± 12.5; Appendix A). However, after omitting one swab sample that revealed exceptionally low amounts of DNA (0.02 ng/mL ± 0.03 ng/mL), DNA concentrations did not significantly differ (two-tailed paired t-test, *p* = 0.0996, *t* = 1.785, df = 12). 

### 3.2. SNP Numbers, Coverage, and Correlations

The mean number of sequencing reads (mean ± 1 S.D.) generated per individual from tissue samples (1,608,863 ± 307,130) was 2.1× more than the number of reads generated from swab samples (518,985 ± 238,072), and this was a highly significant difference (Appendix A; *t* = 9.693, *p* < 0.001). However, DNA concentration was not predictive of the number of raw reads that were ultimately generated, regardless of the DNA sampling technique that was utilized (ANCOVA; *t* = 1.2003, *p* = 0.284).

When both datasets were aligned to the control reference, tissue samples produced more SNPs for each individual than swab samples, but at lower coverage. For the tissue dataset, there were 36,764 SNPs identified, each with a mean coverage of 3.64 ± 0.74 reads per SNP, whereas for the swab dataset, an order of magnitude fewer SNPs (*n* = 3,138) were identified with an average coverage of 4.29 ± 1.94 reads per SNP (Appendix A). For the tissue-only assembly (i.e., tissue samples aligned to tissue-only reference), mean coverage of the 8,111 SNPs was 6.79 ± 1.48, whereas for the swab-only assembly (i.e., swab samples aligned to swab-only reference), mean coverage for the 565 SNPs was dramatically higher at 22.64 ± 10.5 (Appendix A).

### 3.3. Comparing PCA Plots

None of the four PCA plots that were generated revealed genetic structure among individuals sampled from the two separate locations along the Sabine River (Appendix A). When each sample type was aligned to its respective assembly (i.e., tissue data aligned to the tissue-only reference assembly and swab data aligned to the swab-only reference), there was an exceptionally high correlation between the two PCA plots (Procrustes R = 0.982, Figure 1A). Similarly, when the tissue and swab data were separately aligned to the control reference assembly, the PCAs were also highly correlated (Procrustes R = 0.975, Figure 1B). 

## 4. Discussion

Identifying DNA collection methods that minimize mortality and disruption to organisms is important when the taxa being studied are rare or are of high conservation priority [6,7,8,9,10]. Our results demonstrated that both swabbing and tissue biopsy collection methods produce quality DNA in the unionid mussel *F. askewi.* Of the 14 individuals sampled, there was one notable exception where the DNA concentration of a single swabbed sample was near zero (two of three DNA quantification measurements on the qubit system were “zero” and one measurement was at the lowest possible reading of 0.1 ng/µL). As we did not re-extract the sample, it is unknown whether this was a failure of the swab collection method itself or whether this was simply a failed DNA extraction. It is important to note that, except for the potentially failed extraction, should the quantity of DNA collected be important for the study being conducted, swabs produce approximately 20 percent less DNA than tissue biopsies (though this was not significantly less in the current study [*p* = 0.097]), and this may need to be taken under advisement should other experimental protocols require higher concentrations of DNA.

Both methods of DNA collection studied here have proven to be effective at producing high-quality DNA for Sanger sequencing in other species of freshwater mussels [14,15]. In the current study, GBS sequencing of tissue samples generated more than double the number of raw reads per individual compared with those of swab samples. The tissue samples also generated more SNPs per individual, regardless of whether the reads were aligned to the reference assembly or the tissue-only de novo assembly, but the mean read depth of each tissue-based SNP was much lower than that of the mean read depth for the swab-based samples. Regardless of the sampling method, however, the Procrustes analysis revealed nearly identical relationships among all of the included samples. Therefore, the less invasive swab method of DNA collection is likely to be the more acceptable sampling method for performing population genomic work in unionid mussels. However, it should be kept in mind that these results should be interpreted within the context of their sampling distribution. There was no detectable population genetic structure among the individuals collected at the two different locales within the Sabine River (separated by ~97 river km). Thus, the genetic diversity of the individuals is likely lower than what might be expected in individuals collected from a larger sample size and across a wider sampling distribution. It is therefore possible that the extra genomic data provided from tissue biopsy samples may provide a substantial benefit when it comes to identifying any large-scale or fine-scale patterns of structure that might exist in more genetically diverse collections.

We found that visceral swabbing and tissue biopsies revealed similar patterns of genetic variation among individual samples of *F. askewi* when performing GBS, and such methods have the potential to be transferable to other mussel systems as well [16]. Future researchers should consider the likely advantages and disadvantages of both methods. While tissue samples provided substantially more genomic coverage (i.e., more total SNPs), the mean sequence coverage per SNP was higher for the viscera swabbing method, with a concomitant increase in the accuracy of calculating genotype probabilities. Researchers should therefore not only keep in mind these potential tradeoffs, but also take into consideration the degree to which different DNA collection methods have the potential to disrupt growth and survivorship of these species. 

## Figures and Tables

**Figure 1 genes-14-01197-f001:**
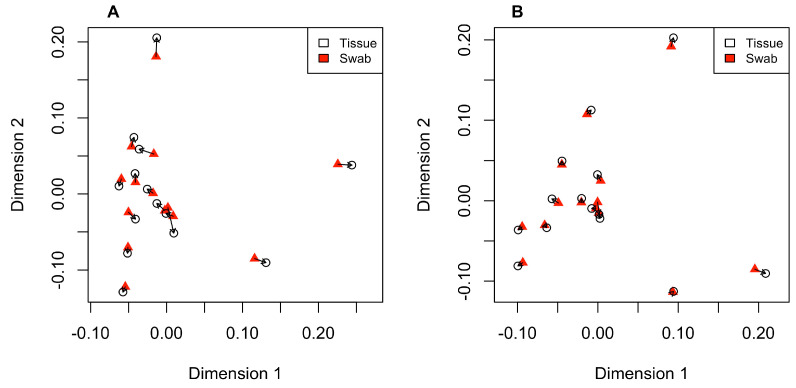
Procrustes plots showing differences between the two independent PCAs when samples were aligned to their own reference assemblies (**A**) and when they were aligned to the control reference (**B**). The open black circles are individual tissue samples, red triangles are individual swab samples, and arrows point to the relative change of positioning between the two PCAs.

## Data Availability

Genotype probabilities, coverage, and parsed reads for all individual samples are uploaded to the following: https://datadryad.org/stash/share/3MrJXCZsYfm0qvQhTmtR-RmJxHbE_bYCuEJ3h-vP_Bc (accessed on 29 May 2023).

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
