# Peer review of "A Comparison of Non-Destructive Visceral Swab and Tissue Biopsy Sampling Methods for Genotyping-by-Sequencing in the Freshwater Mussel Fusconaia askewi"

_genes, 2023, doi:10.3390/genes14061197_

Round 1

Reviewer 1 Report

After carefully reviewed the manuscript entitled "A comparison of non-destructive visceral swab and tissue biopsy sampling methods for genotyping-by-sequencing in the freshwater mussel Fusconaia askewi" by Harrison et al. I am pleased to report that this paper is well-written and the material and methods are complete. The results and discussion are clear and well-explained, making it an interesting read for the readership of the journal "Genes".

The study compares two DNA sampling methods, namely visceral swabbing and tissue biopsies, for generating genotyping-by-sequencing (GBS) data for the freshwater mussel Fusconaia askewi. The study finds that both methods generate quality sequence data, with tissue biopsies producing significantly higher DNA concentrations and larger numbers of reads than swabs. However, swabbing produced greater sequence depth, while tissue biopsies revealed greater coverage across the genome at lower sequence depth. Despite these differences, patterns of genomic variation were similar regardless of the sampling method, suggesting that swabbing is a viable option for producing quality GBS data in these organisms.

Overall, I find the manuscript to be well-executed and informative. The authors have provided a thorough and insightful analysis of the two sampling methods, highlighting their respective advantages and limitations. I believe this study will be of interest to the readership of "Genes" and recommend its publication in the journal.

Few comments:

-       Throughout the manuscript: Please avoid capital letter for “Unionid” or the common name of species (ex. texas pigtoe and not Texas Pigotoe)

-       Line 40: add “the family” before “Unionidae”

-       Line 42: add some of the main conservation issues for this taxon

-       Line 43: “to reduce” instead of “that reduce”.

-       Line 47: what do you mean with “growth characteristics?”

-       Line 50: Please rephrase, you mention 4 methods, two were not as good as for DNA extraction success and you jump to the fact that visceral swabbing is better even than survivorship is 100% with all methods, something is missing.

-       Line 55: what does it means less disruption.

Author Response

-       Throughout the manuscript: Please avoid capital letter for “Unionid” or the common name of species (ex. texas pigtoe and not Texas Pigotoe)

Thank you for the suggestion. “Texas” is a proper place-name, so this cannot be made lowercase, but we have replaced “P” with “p” in “pigtoe” throughout the manuscript.

-       Line 40: add “the family” before “Unionidae”

This has been corrected throughout the manuscript.

-       Line 42: add some of the main conservation issues for this taxon

The following was added to address this comment:

In North America, the family Unionidae is a speciose group of freshwater mussels consisting of both widely distributed and locally endemic species that exhibit varying conservation statuses [11-13]. Dam construction, channel modification, siltation, and the introduction of nonnative freshwater mollusks have all contributed to population declines in many of the species within this diverse group [13].

-       Line 43: “to reduce” instead of “that reduce”.

This was corrected in the manuscript.

-       Line 47: what do you mean with “growth characteristics?”

This was changed to “growth rates.”

-       Line 50: Please rephrase, you mention 4 methods, two were not as good as for DNA extraction success and you jump to the fact that visceral swabbing is better even than survivorship is 100% with all methods, something is missing.

We agree this was confusing. We re-wrote this to read as: “The authors additionally evaluated individual survivorship (after 128 days) across each of the sampling techniques and found that survivorship was 100% for all methods. Because the DNA quantity and quality of swab samples were not only reliable but also less invasive than tissue biopsies, the authors concluded that visceral swabbing was the most appropriate method for tissue collection in M. margaritifera.”

-       Line 55: what does it mean… “less disruption?”

We replaced “less disruption” with “physical injury.”

Reviewer 2 Report

The article "A comparison of non-destructive visceral swab and tissue 2 biopsy sampling methods for genotyping-by-sequencing in the 3 freshwater mussel Fusconaia askewi" from Harrison et al., is a highly didactic piece of research. The title perfectly fits the content of the article and the conclusions are supported by the results. The discussion is very nicely organized and presents the pros and cons of the different methods and their applicability depending on the biological question upstream the experimental design.

The results of this article are a valuable tool/resource for future research projects that will have to select the most efficient and less distructive method to produce high quality population/evolutionary genomic studies. 

However, the article is mainly focused on the comparison of different methods while the journal " welcomes studies using any kind of genetic markers, as long as they provide interesting novel insights into the biology of the organism or into more general population genetics and evolutionary questions.  Studies based on genomic data are particularly welcome". 

Accordignly, I am not sure the article in its current form fits the aims and scopes of the journal. The article should then be revised in order to better highlight the relevant results and take home messages in line with the aims and scopes of the journal.

Author Response

Accordignly, I am not sure the article in its current form fits the aims and scopes of the journal. The article should then be revised in order to better highlight the relevant results and take home messages in line with the aims and scopes of the journal.

We thank the reviewer for the nice comments.

We would note that Reviewer 2 quotes the “Population and Evolutionary Genetics and Genomics” SECTION of the journal, not the scope of the journal overall.

For instance, the manuscript would fit nicely within the “Technologies and Resources for Genetics” section of the journal, as the reviewer is right to note that the primary objectives were to compare sampling methods. However, we feel that this topic is broadly interesting to Population Biologists as a whole and would be more visible to them in the current section (i.e. we are comparing sampling methods specifically for use in examining population structure). We have added a sentence to showcase more the results of the population genetic studies: “There was no detectable population genetic structure among the individuals collected at the two different locales within the Sabine River (separated by ~97 river km),” and would prefer that the article remain in the current section. However, if the editor sees fit to place it into the “Technologies and Resources for Genetics” section, we would find that an acceptable alternative.